# A wireless W-band 3D-printed temperature sensor based on a three-dimensional photonic crystal operating beyond 1000 °C

Jesús Sánchez-Pastor [1] ✉, Petr Kaděra[2], Masoud Sakaki [3], Rolf Jakoby[1], Jaroslav Lacik[2], Niels Benson[3] & Alejandro Jiménez-Sáez [1] ✉

In addressing sensing in harsh and dynamic environments, there are no available millimeter-wave chipless and wireless sensors capable of continuous operation at extremely high temperatures. Here we present a fully dielectric wireless temperature sensor capable of operating beyond 1000 °C. The sensor uses high-Q cavities embedded within a three-dimensional photonic crystal resonating at 83.5 GHz and 85.5 GHz, and a flattened Luneburg lens enhances its readout range. The sensor is additively manufactured using Lithography-based Ceramic Manufacturing in Alumina ($Al_2O_3$). Despite the clutter, its frequency-coded response remains detectable from outside the furnace at 50 cm and at temperatures up to 1200 °C. It is observed that the resonance frequencies shift with temperature. This shift is linked to a change in the dielectric properties of $Al_2O_3$, which are estimated up to 1200 °C and show good agreement with literature values. The sensor is thus highly suitable for millimeter-wave applications in dynamic, cluttered, and high-temperature environments.

The German Fire Protection Association sets a maximum temperature of 1200 °C on normatively regulated design fires employed in the testing of methods for fire protection[1]. Furthermore, this is also the maximum in-room temperature that was measured in a study of the burning of a small house[2]. Moreover, fires can abruptly increase their temperature through phenomena such as flashovers, which typically reach temperatures in the range of 1000 °C[3,4]. In such complex situations, some works have proposed the employment of augmented reality technologies to aid the task of rescue services[5]. Some examples include the employment of virtual maps[6,7], early assistance to find evacuation routes through unmanned aerial vehicles[8], combining thermal imaging and deep learning for early detection of persons in heavy smoke scenarios[9], or monitoring a fire post-ignition state in real time[10].

The aforementioned systems can potentially be enhanced with technology operating in the sub-THz frequency range (100 GHz to 300 GHz) and in the THz spectrum. One key advantage of this frequency range is that it presents a "sweet spot" between the microwave range deep penetration capabilities through non-metallic materials and the fine spatial and time resolution of the optical frequency range. In other words, sensing and

mapping of the environment can be achieved through obstacles such as smoke[11] or walls[12]. For example, the health status of persons within the building could be ascertained, as some works have determined that it is possible to determine whether a person is under physiological stress[13,14] and measure its thoracic movement[15]. Moreover, the gases present in the fire can be identified[16,17].

In indoor self-localization systems, which provide positioning information to rescue robots or UAVs, this frequency range also shows the potential to achieve millimeter accuracy. To achieve so, an unsynchronized and low-complexity infrastructure has been proposed in the form of deploying positioning anchors at fixed positions in the indoor environment[18,19]. These anchors are implemented by mass-deployed sub-THz passive chipless RFID tags, which are designed with customized frequency signatures to distinguish between them[20,21]. They have the advantage of being energy-autonomous, i.e., they do not employ batteries but employ the interrogation wave that reaches them to operate. In this regard, millimeter positioning accuracy has been demonstrated in the W-band (75 GHz to 110 GHz) based on this approach[22]. Most recently, the term "cooperative

[1]Institute of Microwave Engineering and Photonics, Technical University of Darmstadt, Merckstraße 25, Darmstadt, 64283 Hessen, Germany. [2]Department of Radio Electronics, Brno University of Technology, Technicka 3082/12, Brno, 61600 Brno-město, Czech Republic. [3]Institute of Technology for Nanostructures, University of Duisburg-Essen, Bismarckstraße 81, Duisburg, D-47057 North Rhine-Westphalia, Germany. ✉e-mail: jesus.sanchez@tu-darmstadt.de; alejandro.jimenez_saez@tu-darmstadt.de

chipless RFID tags" was employed to refer to the extraction of environmental data, on top of position, from the tags[23]. In other words, to turn them into sensors. However, there are currently no available options for sub-THz tags able to operate in the harsh environments expected from indoor fires.

Available temperature sensor concepts operating at lower frequencies (around 10 GHz and over 1000 ˚C) present very limited wireless readout ranges, below 10 cm[24–29]. Some research has tried to address these short-range issues by employing the backscattered radiation of spherical dielectric resonators[30,31], achieving readout ranges of more than 1 m. However, the decrease in geometrical size (and hence in readout range) associated with an increased frequency limits their usefulness at the sub-THz range.

In light of the above, it is useful to consider a related sub-THz research area, namely the characterization of dielectric materials, to understand the current state-of-the-art of operation at high temperatures in this frequency range. Two key techniques stand out for this study: wireless measurements[32,33] and cavity-based spectroscopy[34,35]. High-temperature wireless characterization systems build upon the proposal of Varadan et al.[36] Here, a sample material is placed in a furnace with two openings, and the transmitted wave through the sample is used to extract its broadband dielectric properties. This method has been successfully scaled up to the W-band and 1200 ˚C by other authors[37–39]. Conversely, cavity-based methods use a rod of the test material and analyze changes in the cavity's frequency response to estimate both the material's relative dielectric permittivity, $\varepsilon_r$, and dielectric losses, $\tan(\delta)$[40,41].

In this work, a wireless W-band fully dielectric temperature sensor is designed inspired by cavity spectroscopy, as a first step towards sub-THz chipless RFID tags for harsh environments. It is 3D-printed in Alumina ($Al_2O_3$) via Lithography-based Ceramics Manufacturing. $Al_2O_3$ is chosen due to being a ceramic (inherently suited for harsh environments), the possibility of employing 3D printing to aid the implementation of complex structures and presenting extremely low dielectric losses in the W-band[42]. The sensing is implemented with two reflective cavities embedded within a 3D photonic crystal (PhC) lattice, while their low readout range is enhanced by adding a flattened lens antenna. The main contributions of this concept are (i) foregoing the employment of metallic cavities in favor of fully dielectric ones, (ii) isolating the cavities' responses from their surrounding environment and (iii) achieving a wireless readout range large enough to detect the sensor inside a furnace ($\approx 0.5$ m) for characterization purposes. The operation of the sensor is described up to 1200 ˚C, which is the maximum operating temperature of the employed furnace.

## Results
### Sensor design
**Operation principle.** An indoor environment in a rescue situation is a highly dynamic and cluttered environment, where the echoes from the environment might mask the response of RFID tags distributed at fixed positions. Thus, some type of clutter suppression is required. An electronics-free solution is to employ high Q-factor cavities to encode the sensor information on its backscattered wave[20]. The long ringing response of the sensor (due to the high-Q cavities) allows for isolating the sensor's response in the time domain after the clutter echoes have decayed so that more robust readouts are achieved in the form of peaks at the $f_{res}$ of the cavities. A more detailed description of this principle is presented in the Supplementary Note S1.

An illustration of the proposed sensor is presented in Fig. 1. In this diagram, a reader transmits a rectangular signal in the frequency domain, which reaches the sensor and is directed through a flattened lens. The lens focuses the interrogation signal into a frequency-coded reflective layer, where two high-Q cavities have been implemented and are responsible for the sensor's identification and sensing. These two high-Q cavities are excited and re-radiate at their resonance frequency, $f_{res}$ towards the reader. Hence, the reader receives a peak in the frequency domain, corresponding to the high-Q cavities' responses. Since their $f_{res}$ can be set during the design process, different $f_{res}$ can be implemented to achieve independent sensors, therefore the name of the frequency-coded reflective layer. Do note that the

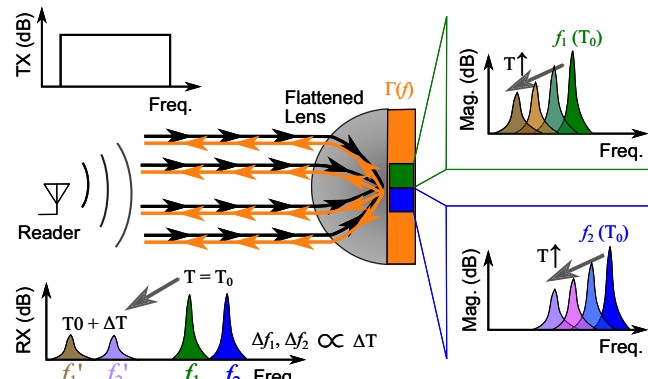

**Fig. 1 | Operating principle of the temperature sensor.** Two cavities (green and blue) are embedded within a reflective layer and a lens is used to increase the readout range.

represented responses assume that both clutter and the sensor's structural mode are not considered so that only the peaks corresponding to the cavities are present in the received signal.

A variation in the ambient temperature results in a change in the dielectric permittivity of the sensor's material. For example, if the material presents an increase of its $\varepsilon_r$ with temperature, then the cavities' responses shift towards lower frequencies. By knowing the sensor's response, the temperature can be calculated from the detected $f_{res}$. Furthermore, in the case that the $\tan(\delta)$ of material increases, the resonance bandwidth broadens and its peak magnitude decreases. This showcases the importance of employing a sensor material that provides sensitivity, i.e., high $\varepsilon_r$ variability with temperature while keeping the $\tan(\delta)$ low enough that the resonance peaks can be detected.

**Photonic crystal-based frequency-coded reflective layer.** The periodic alternation of two dielectric materials generates so-called electromagnetic bandgaps. Due to their interest and similarities to the propagation of an electron in a crystalline material, such structures receive the name of photonic crystals (PhCs). Having a bandgap means that no propagation is possible in the material so that incoming waves of certain frequency bands are reflected. Due to this property, PhCs are often used to surround resonant cavities, achieving extremely high Q-factors[43–45]. PhCs can be classified into one-dimensional, two-dimensional, or three-dimensional (3D) categories based on the periodicity of their conforming elements. This paper employs 3D PhCs, which present bandgaps in all three dimensions of space[46–48].

In light of the above, 3D PhCs are suitable as elements of a temperature sensor. On the one hand, the integration of resonant cavities within the 3D PhC is a simple process, achieved by altering the PhC lattice locally. On the other hand, the 3D bandgap mechanically and electrically isolates the high-Q cavities from the environment, which might affect their $f_{res}$ or loaded Q-factor, $Q_l$. This effectively reduces cross-sensitivity in real, dynamic environments. It must be pointed out that other fully dielectric metasurfaces are also capable of implementing high-Q resonances[49–52]. Thus, the PhC could be potentially replaced by them, provided that they present similar capabilities in terms of confining the cavities' responses in such a way that they are not altered by their surrounding environment.

Figure 2a shows an excerpt of a 3D PhC operating as a frequency-coded retroreflective layer. It is based on the design discussed by Povinelli et al.[53], which we have implemented in $Al_2O_3$. To operate within the W-band, the lattice constant, $a$, is set to 1.56 mm. The implementation of two cavities (point defects) allows for the design of multiple frequency codes to distinguish between sensors in a multi-deployment scenario, by assigning a specific separation between the resonance frequency of the cavities. It also has the further benefit of decreasing uncertainty in temperature estimation, as recommended by Hakki and Colemann[34] for dielectric characterization

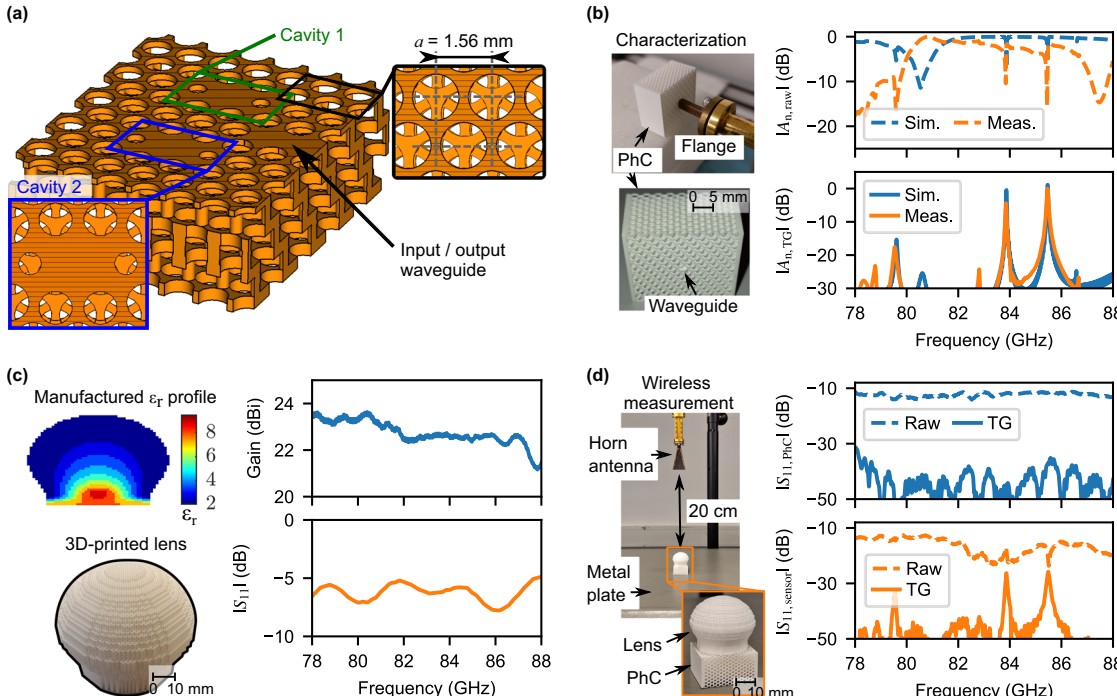

**Fig. 2 | Building blocks of the PhC structure. a** Cross-section of the 3D-PhC with marked lattice constant $a$. The design parameters for the PhC and cavities are provided in the Supplementary Note S2. **b** Normalized comparison between simulated and measured $S_{11}$, where TG = time-gated signal. For a simulated $\varepsilon_r$ = 8.53, the relative differences between simulated and measured $f_{res}$ are of 0.03 %

and 0.00 % for the lower and higher frequency cavities, respectively. **c** Measured gain and $S_{11}$ of the manufactured flattened lens. The lens's implemented gradient index is presented in the inset. **d** Enhanced readout at 20 cm by employing a lens. For all time-gated results in this image, the rectangular time window spans between 2.5 ns and 27.5 ns. PhC photonic crystal, TG time-gating.

purposes. The excitation of the cavities, as well as the backscattering of their responses towards the reader, is achieved by implementing a single input/output dielectric waveguide.

Figure 2b presents both the simulated and measured reflection coefficient, $S_{11}$, of the PhC-based structure. These results have been normalized to facilitate comparison. While the two notches corresponding to each cavity are discernible in the unprocessed responses, this is a situation that might not occur in real, highly cluttered, environments. Consequently, the cavities' high-Q factor can be effectively utilized to isolate their frequency response, which arises as peaks in the frequency spectrum. This is achievable because the backscattered response from the cavities tends to persist amidst environmental scatterers. To isolate the resonance frequencies of the cavities, a rectangular time-domain window is applied from 2.5 ns to 27.5 ns. As a result, the previous minima now appear as two peaks, one per cavity, at their $f_{res}$ of 83.5 GHz and 85.5 GHz, and can be distinctly distinguished from the rest of their surrounding frequency spectrum.

**Flattened lens for enhanced wireless readout.** The small aperture size of the PhC's dielectric waveguide implies that the cavities present a low backscattered power, even if a rod antenna is integrated as proposed in ref. 54. For this reason, the PhC's dielectric waveguide is not matched to the air, but to a 3D-printed gradient index lens antenna. This lens is added to increase both the power that reaches the cavities and the power reflected towards the reader, improving the readout range[55,56]. The lens antenna is flattened, so it can be placed on top of the PhC, with its center corresponding to where the input waveguide is located.

The details of the lens's design employed in this work were presented by Kaděra et al.[57], where it is designed for operation at 40 GHz, following effective medium theory and quasi-conformal transformation optics. In practice, the boundaries of the effective medium theory used in that publication, unit cell period $p < 0.1\lambda_0$ are a conservative estimation for fully dielectric structures. Measurements carried out in the scope of this work show that the lens can be employed up to approximately 98 GHz[58].

Compared to our previous work, we have added a matching layer to the bottom of the lens to match the end of the lens to the input of the PhC waveguide, $\varepsilon_{r,eff}$ = 7.62, instead of to air, to maximize power transfer between the lens and the PhC cavities. This applies in both directions (receive, backscatter) due to the Lorentz reciprocity theorem. Figure 2c shows the measured gain and $S_{11}$ of the designed lens coupled to an open rectangular waveguide flange. In this configuration, the lens antenna presents a gain of 22.5 dBi at the cavities' $f_{res}$, while the $S_{11}$ is of the order of −7 dB. The large reflection is due to the mismatch between the hollow waveguide ($\varepsilon_r$ = 1) and the lens bottom, which is designed for the PhC's dielectric waveguide.

To illustrate the enhanced readout range, Fig. 2d displays two measurements conducted both without and with the inclusion of a lens, taken at a distance of 20 cm from the transmitting/receiving antenna. A clear observation can be made: with the incorporation of the lens, there is a notable enhancement in received signal power. While the lack of the lens meant that the cavities' responses were not appreciable (hidden by noise), its addition allowed them to stand out from the rest of the spectrum by approximately 20 dB. The measured radar cross-section of the sensor at its resonance peaks (i.e., after the cavities's responses are isolated via time-gating) is −20 dB m$^2$, as described in the "Methods" section.

**Temperature characterization**

A schematic of the measurement setup is shown in Fig. 3a, where the device under test (DUT) corresponds to the sensor resulting from the combination of the 3D PhC and the lens. The separation between the measuring antenna and DUT is set to 50 cm. The employed furnace can be considered as a highly cluttered environment, as it is enclosed by metal on all sides, except for the top part. The top is closed by a thin layer of concrete combined with a thick isolating bag made of ceramic fiber, which prevents outwards heat radiation while allowing for incoming signals to enter the furnace and therefore excite the cavities. Thus, multiple reflections are expected inside the furnace.

The measured $S_{11}$ for the time domain is shown in Fig. 3b, where the reflections caused by the obstacles in the signal's path are appreciable as

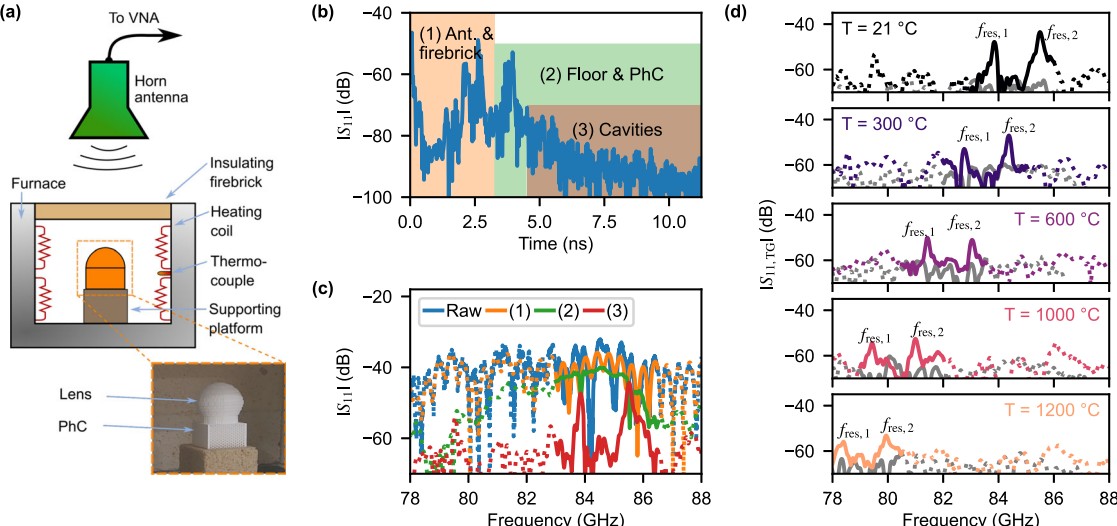

**Fig. 3 | High-temperature measurement setup and results. a** Schematic of the setup. For more details and photos of it, please refer to the "Methods" section and Supplementary Note S3. **b** Measured backscattered time-domain response, where the different environmental reflections and cavity responses are marked by the colored rectangles. **c** Measured frequency-domain response for the colored windows marked in the time domain. The two peaks at 83.85 GHz and 85.50 GHz correspond to the cavities' response and present a magnitude approximately 15 dB higher than the rest of the spectral response. **d** Backscattered frequency responses for different temperatures, where the gray lines correspond to the response of the empty furnace. The time window spans between 4.5 ns to 11.75 ns for all results. PhC photonic crystal, TG time-gating, $f_{res}$: resonance frequency; $T$ temperature.

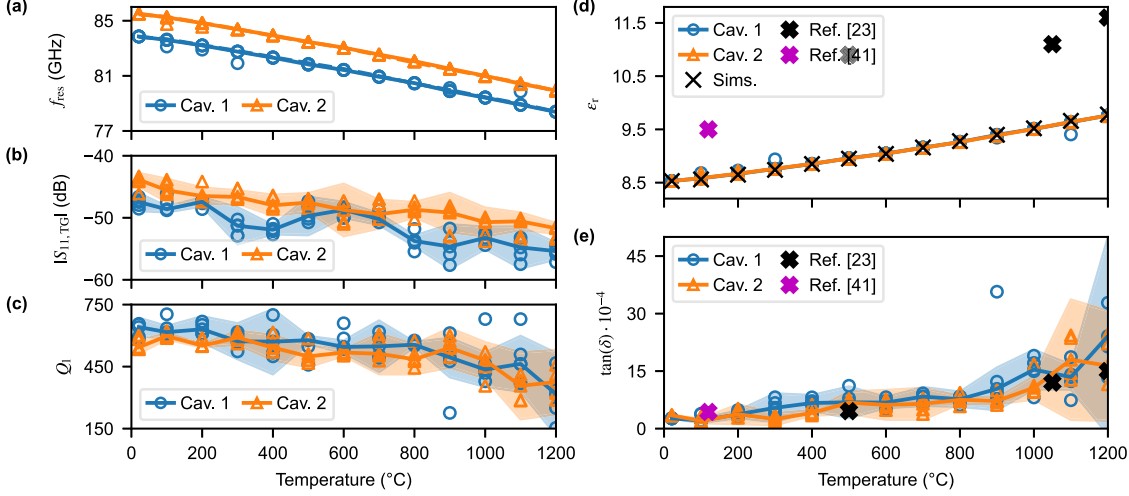

**Fig. 4 | Measured results and extracted dielectric properties of Al₂O₃ regarding temperature.** Variations in the measurements are considered to follow a normal distribution. Therefore, the markers in plots (**a**–**c**) represent the result for each measurement, the solid lines the median, and the shadowed area the 95% interval.

Five measurements are taken for each temperature ($N = 5$). The results for the first cavity are marked in blue and for the second cavity in orange. **a** $f_{res}$ for the two cavities, compared to the simulated response (dashed lines). **b** $Q_l$ of each cavity. **c** $S_{11}(f_{res})$. **d, e** extracted $\varepsilon_r$ and $\tan(\delta)$ of Al₂O₃, respectively.

multiple peaks, which are classified depending on which part of the furnace that reflection took place. Moreover, it can be appreciated that the signal presents a slow decay rate after approximately 3 ns, corresponding to the cavities' response. Their corresponding time-gated frequency-domain signals are displayed in Fig. 3c. It is noticeable that the raw signal does not present any notch corresponding to the cavities, owing to the large reflection caused by the measuring antenna and the furnace's lid. However, the notches are appreciable when this part of the signal is filtered out, with the cavities' peaks arising after further removing the structural reflection of the PhC structure. In light of the above, it is appreciable that the sensor can be wirelessly read out from distances of 50 cm and larger.

Measurements for different temperatures up to 1200 °C are exhibited in Fig. 3d. It is noticeable, that (i) the shift in $f_{res}$ is towards lower frequencies, (ii) the $S_{11}$ peak at resonance decreases its magnitude, and (iii) the resonance

peaks broaden for increasing temperatures. The frequency shift is related to both the thermal expansion and the variation in the $\varepsilon_r$ of Al₂O₃ over temperature, while the lower peak magnitude and its broadening are a consequence of increased dielectric losses. Regardless, it is appreciable that the peaks corresponding to the two cavities remain above the surrounding frequency spectrum, enabling both the identification of the sensor and the estimation of its temperature.

In Fig. 4a, the measured $f_{res}$ for each cavity is depicted compared to simulations that consider both the change in $\varepsilon_r$ of Al₂O₃ over temperature and the thermal expansion of the cavity. It is noticeable that the $f_{res}$ shifts almost linearly towards lower frequencies, i.e., the $\varepsilon_r$ of Al₂O₃ increases linearly with increasing temperature, with a median $f_{res}$ shift of $-57$ ppm·K$^{-1}$. Furthermore, in Fig. 4b, the measured magnitude of the $|S_{11}|$ at resonance (after time-gating) is displayed, presenting a lower magnitude

for the first cavity than for the second. This is accompanied by a larger overall standard deviation in the estimation of the $Q_l$, shown in Fig. 4(c). Although the resonance peaks' magnitudes remain above the measurement equipment noise level ($\sim -60$ dB) for all temperatures, the $-3$ dB points employed to calculate the $Q_l$ are very close to it, which introduces an uncertainty in the measured results. Therefore, five ($N = 5$) independent measurements are taken to properly assess the sensor's behavior. It is also noticeable that the $Q_l$ sharply decreases after 900 °C, which indicates an abrupt increase in the material losses.

Due to employing the same material for the lens and the high-Q cavities, as well as the feasible wireless readout of the sensor, a characterization of the material properties is possible. The process employed to obtain the dielectric properties, $\varepsilon_r$ and $\tan(\delta)$, of $Al_2O_3$ over temperature is explained in the "Methods" section, while the extracted results are presented in Fig. 4d, e, respectively.

As predicted from the resonance frequency shift, the $\varepsilon_r$ increases with increasing temperature, which matches the results reported by Katz[41] at similar frequencies of this work. In Fig. 4d, the estimated $\varepsilon_r$ is presented employing two methods. Firstly, by directly calculating its change by employing the frequency shift (denoted by Meas.). Secondly, by considering the thermal expansion of $Al_2O_3$ and matching measured and simulated $f_{res}$ (represented by Sims.) by sweeping over $\varepsilon_{r,sim}$. Due to the low thermal expansion of $Al_2O_3$, it can be appreciated that both approaches extract almost the same $\varepsilon_r$, and therefore the frequency shift is weakly dependent on thermal expansion up to 1200 °C. Two of the main effects responsible for the increase of $\varepsilon_r$ are an increased molecular motion, due to the high temperatures, which can result in enhanced dipole moments, and the ionization of defects present in the grain boundaries of the material, resulting in an increased number of charge carriers. These effects increase the effective polarizability of the material, and hence its $\varepsilon_r$.

The extracted $\tan(\delta)$ is depicted in Fig. 4e, where it is noticeable that the losses of the material increase with temperature. The value extracted for 120 °C is in good agreement with the result obtained by Jiménez-Sáez[54]. The increase in dielectric losses at higher temperatures corresponds to the characterization presented by Katz[41], particularly for the second cavity, due to its higher backscattered power. For the first one, a larger degree of deviation is present, due to the aforementioned proximity to the noise floor. The increase in losses is also related to the two effects that justify the increase in $\varepsilon_r$: greater molecular motion results in larger friction and collisions between molecules, which increases dielectric losses. Moreover, ionic conduction increases energy dissipation within the material. In fact, Katz[41] and Ho[59] propose this mechanism as the main factor for the large loss increase above 900 °C–1000 °C.

### Limitations and discussion

The sensor presents several limitations and constraints that should be considered, in terms of readout range, re-design at different frequencies, temperature sensing in real settings, uncertainty in $\varepsilon_r$, and influence of changes in the surrounding ambiance.

First, the measurement distance in this work has been set to 0.5 m, which is chosen to detect the resonance peaks at 1200 °C, one of the reasons is its relatively low radar cross-section, of $-20$ dB m². However, this does not imply that the sensor is limited to this short distance. Rather, different parameters of the monostatic setup, particularly the reader's antenna gain, can be modified so that the sensor can be detected even up to 18 m, as described in Supplementary Discussion S4, which enables its employment in indoor settings.

Second, the overall size of the tag presented in this work is large (even more after adding the lens), in a field where usually smaller structures are preferred, as thinner structures lead to more homogenous and quicker heating or cooling, decreasing the possibility of temperature gradient in the structure that might lead to false measurements. In this regard, our structure is porous, which partly compensates for this effect. The time constant, $\tau_t$, of the sensor is highly dependent on its surrounding conditions (convection air currents, metallic pads, or materials to increase heat transfer...). Depending

on the boundary conditions assumed, $\tau_t$ ranges between 41 s and 2 min (see Supplementary Discussion S5). These response times could be further improved if air convection through the 3D PhC grid is achieved, an effect that requires further investigation.

Third, the sensor was measured between 21 °C and 1200 °C. Considering that the maximum working temperature of LCM-printed $Al_2O_3$ is 1650 °C, it can potentially be employed up to this temperature, provided that the increase in dielectric losses allows for its wireless readout. For higher temperatures, the structure softens and loses its functionality. Moreover, the lens and PhC are manufactured separately. For its employment in real environment settings, the monolithic fabrication of the sensor is necessary, for designing a mechanical support structure for the integration of the two parts.

Finally, the employment of a 3D PhC, as aforementioned, reduces cross-sensitivity against the surrounding ambiance. That is, considering its potential application for indoor localization, the sensor could be installed within walls or roofs, with the PhC part inside and the lens protruding from them. Being surrounded by concrete, plaster, steel or other common building materials does not alter the sensor's response. However, this situation changes if it is submerged in an environment different from the air. A decrease in the $\varepsilon_r$ contrast between $Al_2O_3$ and its surrounding media leads to a weakening of the bandgap and a decrease in the $Q_l$ of the cavities, as well as their shift towards lower frequencies. Further, very lossy environments can lead to the broadening and combination of the two cavities' responses. This decrease in $Q_l$ shortens the cavities' response and limits the sensor's robustness against clutter.

### Conclusion

In this work, a fully dielectric, chipless temperature sensor operating at W-band frequencies is presented, which is 3D-printed in $Al_2O_3$ via Lithography-based Ceramic Manufacturing. Its operating principle relies on a variation in its material properties, $\varepsilon_r$ and $\tan(\delta)$, over temperature, which result in a distinguishable shift on the sensor's frequency response.

In this regard, two high-Q cavities are embedded within a 3D PhC lattice, and a flattened lens is added to achieve wireless performance, which enables the sensor's readout inside a furnace at a distance of 50 cm and up to 1200 °C. This range can be further extended by considering the factors influencing the readout range in monostatic radar systems, such as larger reader antenna's gain, higher transmitted power, or averaging through multiple measurements. The high-Q factor of the cavities allows for easy distinction of the resonance frequency peaks from undesired environmental scatterers by time-gating techniques. Table 1 presents a comparison between wireless temperature sensors operating in the microwave regime at 1000 °C and above. It is noticeable that the ones based on $Al_2O_3$ present the greater sensitivity to temperature changes, $\tau_f$. Further, most of them feature a very limited readout range below 10 cm (with the notable exception of the work by Kubina[30]) in comparison to the 50 cm that was achieved in this work. Finally, we demonstrated how this concept can be leveraged to extract not only the $\varepsilon_r$, but also the $\tan(\delta)$ over temperature of low-loss dielectric materials at the cavities' resonance frequencies.

As the sensor concept relies on 3D-printed and dielectric structures, it can be potentially extended to higher frequencies, allowing for both accurate sensing and characterization of $\varepsilon_r$ and low $\tan(\delta)$ over temperature at sub-THz and THz frequencies. At these spectrums, inherent ohmic losses and surface roughness play a critical role in sensors that employ metallic components, while both dielectric PhCs, as well as cavities embedded within them, have been shown to be robust against manufacturing effects, preserving the bandgap even for extreme circumstances[60], and achieving high-Q factors[61,62].

### Methods

#### Estimation of the lattice constant

As mentioned in the publication, the design is based on the results presented by Povinelli et.al.[53]. In the publication, they present a band structure diagram

**Table 1 | Wireless chipless microwave resonator-based temperature sensors operating at or beyond 1000 ˚C**

| Refs. | $f_{res}(T = T_0)$ (GHz) | Time-gating | $T_{meas}^{max}$ (˚C) | Material | $\tau_f$ (ppm/K)[a] | $d_{meas}$ (cm) |
|---|---|---|---|---|---|---|
| 24 | 5.2 | Yes | 1000 | Al$_2$O$_3$ | −78 | 3 |
| 25 | 10 | Yes | 1300 | Si4B1 | −45 | 2.5 |
| 26 | 15 | Yes | 1100 | PDC-SiAlCN & platinum | −16 | 1.4 |
| 27 | 10.5 | Yes | 1250 | SiCNO-BN | −26 | - |
| 28 | 13.5 | No | 1100 | PDCs-SiAlCN | −25 | 3.2 |
| 29 | 14 | No | 1000 | Al$_2$O$_3$ & copper | −58 | 5.5 |
| 30 | 3.5 | Yes | 800[b] | Al$_2$O$_3$ | −63 | 150 |
| This work | 85 | Yes | 1200 | Al$_2$O$_3$ | −57 | 50 |

$f_{res}$, resonance frequency, $T$ temperature, $\tau_f$ sensitivity, $d_{meas}$ measurement distance.
[a]Defined as $\tau_f = (1/f_{res,T_0}) \cdot (\Delta f_{res}/\Delta T)$, employing minimum and maximum measurement temperatures.
[b]Added for long-range comparison.

calculated for an $\varepsilon_r$ contrast of 12:1. We estimated a first approximation of the lattice constant, $a$, at 87.5 GHz (center of the W-band), based on their diagram, so that $f \cdot a/c_0 = 0.4$. Afterwards, we employ the scaling properties of the Maxwell equations, by calculating $a$ when the $\varepsilon_r$ is decreased from 12 to 9.5, which was our first consideration of $\varepsilon_r$ for Al$_2$O$_3$[42]. Thus:

$$a = \sqrt{\frac{12}{9.5}} \cdot 0.4 \cdot \frac{c_0}{f} = 1.54 \, \text{mm} \tag{1}$$

The final value reported, $a = 1.56$ mm, is the result of an optimization process in CST Studio Suite to slightly shift the resulting EBG of the 3D PhC.

### Dielectric permittivity and loss tangent estimation
The extraction of the dielectric permittivity, $\varepsilon_r$, and dielectric losses, $\tan(\delta)$, of 3D-printed Alumina (Al$_2$O$_3$) for different temperatures is based on the measured shift of the resonance frequency of the 3D-PhC-based cavities and their Q-factors, respectively.

**Estimation of dielectric permittivity**. The process of estimating $\varepsilon_r$ is separated in obtaining this value for room temperature and for the rest of the considered temperatures.

At room temperature ($T_0 = 21$ ˚C), the structure dimensions are characterized via an optical microscope, to account for variations in the manufacturing process. Then, a simulation of the 3D-PhC is prepared employing those parameters. A sweep in terms of $\varepsilon_r$ is performed until the measured and simulated $f_{res}$ of both cavities match, with an allowed maximum average deviation of 0.05 % between them. The $\varepsilon_r$ for which this condition is fulfilled is considered the $\varepsilon_r$ of the 3D-printed Al$_2$O$_3$ at room temperature.

The heating of the structure causes a shift in the resonance frequency of the cavities, which can be described by the relationship between the resonance frequencies at room temperature and at the targeted temperature, $s_{f_{res}}$:

$$s_{f_{res}}(T) = \left( \frac{f_{res}(T)}{f_{res}(T = T_0)} \right)^{-2} \tag{2}$$

There are two effects that affect this parameter, namely the change in $\varepsilon_r$ with temperature and the thermal expansion of the cavity, such that:

$$s_{f_{res}} = s_{th} \cdot s_{\varepsilon_r} \tag{3}$$

with $s_{th}$ modeling the thermal expansion of the cavity and $s_{\varepsilon_r}$ describing the change in $\varepsilon_r$. From Eq. (3), the $s_{\varepsilon_r}$ can be obtained, and from it the $\varepsilon_r$ for different temperatures estimated, following:

$$\varepsilon_r(T) = \varepsilon_{r,T_0} \cdot s_{\varepsilon_r} \tag{4}$$

The thermal expansion coefficient of Al$_2$O$_3$, $\alpha_{th}$, is reported to be between 7 ppm/K to 8 ppm/K[63]. In our case, we chose an intermediate value, $\alpha_{th} = 7.5$ ppm. The increased size of the structure with temperature can therefore be described by $s_{th}$, such that:

$$s_{th} = 1 + \alpha_{th} \cdot \Delta T \tag{5}$$

where $\Delta T$ is the difference between the measurement temperature and the room temperature. For our measurements, this coefficient has a value of 1.0088425 for the maximum measurement temperature of 1200 ˚C. Thermal expansion enlarges the cavity size for higher temperatures, which contributes to a greater $s_{f_{res}}(T)$ than if the frequency shift was completely caused by a change in $\varepsilon_r$. However, due to the low value of $\alpha_{th}$, its impact is negligible up to 1200 ˚C. In fact, when taking it into account and estimating the $\varepsilon_r$ via matching of the measured and simulated $f_{res}$, the maximum relative deviation in the estimated $\varepsilon_r$ is 0.34 %. Therefore, Eqs. (2) and (4) can be employed directly, neglecting thermal expansion, assuming $s_{\varepsilon_r} = s_{f_{res}}$.

**Extraction of dielectric losses**. Once the $\varepsilon_r(T)$ is determined, the dielectric losses in dependence of temperature, $\tan(\delta(T))$ are estimated by analyzing the Q-factors of the cavities obtained from measurements and simulations of the fabricated structure. The losses can be extracted following that the loaded (measured) Q-factor, $Q_l$, of the cavities is defined as:

$$\frac{1}{Q_l} = \frac{1}{Q_{rc}} + \frac{1}{Q_{diel}} = \frac{1}{Q_r} + \frac{1}{Q_c} + \frac{1}{Q_{diel}} \tag{6}$$

Where $Q_{rc}$ is the radiation-coupling factor and $Q_{diel}$ the material Q-factor, and equal to $1/\tan(\delta)$. $Q_{rc}$ is further subdivided into two Q-factors that can be estimated from simulations of the lossless cavity. $Q_c$ accounts for the coupling of an EM wave to the cavity and $Q_r$ describes its radiation. Assuming that the cavity is symmetric, these two values are extracted from the simulated Q-factors of the retroreflective cavity, combined with its two-port equivalent. This simulation is generated by adding a symmetric input waveguide, which in turn modifies the $Q_c$, while keeping $Q_r$ unchanged. Therefore, the $Q_c$ of the cavities can be estimated as

$$\frac{1}{Q_c} = \frac{1}{Q_{rc}^{2-port}} - \frac{1}{Q_{rc}^{1-port}} \tag{7}$$

Once $Q_c$ is obtained, the calculation of $Q_r$ is straightforward, since:

$$\frac{1}{Q_{rc}^{1-port}} = \frac{1}{Q_r} + \frac{1}{Q_c} \tag{8}$$

Note that, as the structure in which the cavities are implemented is a 3D PhC, $Q_r$ will be at least one order of magnitude larger than the other Q-factors, as the 3D PhC forbids wave propagation through the structure, and

the effect of the dielectric waveguide is described via $Q_c$. Therefore, $1/Q_r$ can be assumed as negligible.

**Lensed structure calibration**. The introduction of a radar cross-section enhancing structure, in this case, a flattened bottom Luneburg lens, affects the $Q_c$ of the structure, owing to the lens acting as a matching structure. The permittivity gradient of the lens varies from $\varepsilon_r \approx 2$ at its forefront, so it is matched to an EM plane wave propagating through air, to the $\varepsilon_r$ of the PhC waveguide present at its bottom, to maximize power transfer to and from the cavities. This results in a decrease of the $Q_c$, observable by a reduction in the $Q_l$ of the structure. The previous method presented to estimate the losses does not consider the effect of the lens, resulting in an overestimation of $Q_c$ which is linked to an underestimation of $Q_{diel}$ and thus higher computed dielectric losses.

To account for the addition of the lens, an equivalence between the lensless and lensed measurements is necessary. It is assumed that the losses of $Al_2O_3$ remain constant at room temperature, independently of whether radar cross-section enhancing structures are employed or not. Therefore, once the $Q_{diel}$ is estimated without a lens, following the previously described procedure, it remains constant. The $Q_c$ of the lensed structure can be calculated as:

$$\frac{1}{Q_c^{w/lens}} = \frac{1}{Q_l^{w/lens}} - \frac{1}{Q_{diel}^{w/o\ lens}} \qquad (9)$$

Once this value is obtained, the relation between $Q_c^{w/o\ lens}$ and $Q_c^{w/lens}$ is employed to obtain the corresponding value to calculate the $Q_{diel}(T)$:

$$\frac{1}{Q_{diel}(T)} = \frac{1}{Q_l(T)} - \frac{1}{Q_{c,corrected}(T)}$$
$$= \frac{1}{Q_l(T)} - \frac{1}{F_{Q_c} \cdot Q_c^{w/o\ lens}(T)} \qquad (10)$$

where $F_{Q_c}$ is the calibration parameter calculated at room temperature to equate the measurements without and with the lens:

$$F_{Q_c} = \frac{Q_c^{w/lens}(T = T_0)}{Q_c^{w/o\ lens}(T = T_0)} \qquad (11)$$

It is assumed that $F_{Q_c}$ is independent of temperature, which is a simplification that contributes to the uncertainty of the computed losses. Ideally, full EM wave simulations that include the effect of the lens should be employed to compute the corresponding $Q_c(T)$, which requires considerable computational resources.

**Fabrication**
The lithography-based ceramics manufacturing (LCM) technology was employed for the realization of the structures. They are created layer-by-layer via DLP-controlled polymerization of a photosensitive slurry (Litha-Lox 360)[64,65]. LithaLox 360 contains 49 vol/% of high-purity $Al_2O_3$ powder as well as 51 vol/% of UV-curable polymers. The utilized printer was a Lithoz CeraFab 7500 with a printing resolution of 25 μm and a UV light source (wavelength: ~450 nm) for slurry polymerization. An illumination energy of 450 mJ/cm²/layer is employed for the printing process.

After the printing process, the structures are washed by dipping them 5 min in LithaSol 20 cleaning fluid, and then ultrasonication for 2 min. This process was repeated 10 times, to ensure that all unpolymerized slurries were removed from the structures.

The cleaned green samples are subjected to a thermal process to convert them into their dense ceramic counterparts. Samples were slowly dried in a laboratory oven. The maximum temperature and duration of the drying step were 140 °C and 6 days, respectively. Afterward, the dried structures were sintered in an electric furnace under an ambient atmosphere. Heating rates were low up to temperatures of 430 °C (i.e., the temperature by which polymer compounds will be fully decomposed into volatile components) to

guarantee the fabrication of crack-free samples. In this study, the structures were sintered at 1600 °C during 4 days.

**Measurement setup**
All measurements are performed with a Keysight Technologies N5222A vector network analyzer (VNA) along with Anritsu 3740A frequency extenders connected to standard 23 dBi W-Band horn antennas.

**Lens characterization**. The gain of the QCTO lens for frontal incidence is characterized by employing the gain-transfer method[66], where the two known antennas are standard 23 dBi W-Band horn antennas.

**Photonic crystal characterization at room temperature**. The PhC is placed at the output of the W-band flange, which has been previously calibrated via a full one-port calibration. The empty room method is not employed due to the large backscattered power by the PhC rendering it unnecessary. The cavities responses are isolated employing a rectangular time window between 2.5 ns to 25 ns.

**Temperature measurements**. The structure is placed inside a furnace whose top part is open and features an isolation layer made of a thin panel of concrete and a thicker bag of ceramic and glass, which allows the EM wave to propagate inside the furnace, as well as exit it. This configuration implies that the measuring antenna has to be placed perpendicular to the floor, which is accomplished via a W-band flexible waveguide. A one-port calibration is performed at the end of the flexible waveguide, to diminish its response in the measurements. It was found that, otherwise, high-magnitude internal reflections within the flexible waveguide masked the cavities' resonances.

The antenna is placed approximately 50 cm above the lens, to fulfill the far field distance criterion at 85 GHz. To prevent any heating of the antenna and flexible waveguide due to a perceivable residual heat that is emitted from the furnace at the highest temperature, an airflow is generated by a fan. Since the airflow could displace the measuring antenna slightly, which results in a loss of calibration accuracy due to phase shifts, the flexible waveguide is fixed in place with customized support.

The furnace is programmed manually to heat up to 1200 °C at approximately 6.5 °C/min up to 600 °C. Then, this value is changed to 3.3 °C/min. The measurements are taken in two phases. First, between 600 °C to 1200 °C, the furnace is set to maintain a constant temperature for 30 min each time that a multiple of 100 °C is reached before taking the measurement. Second, for temperatures between 100 °C to 500 °C, the measurements are taken while the furnace cools down. The temperature of the furnace is estimated by reading the voltage given by a monitoring thermocouple sensor placed on the side of the furnace. The measurements are automatized via MATLAB.

**Radar cross-section measurement**. The radar cross-section ($\sigma$) of the sensor is measured by employing a reference radar target in the same measurement setup and comparing the backscattered power of the two, so that:

$$\frac{\sigma_{sensor}}{\sigma_{reference}} = \frac{|S_{11,sensor}|}{|S_{11,reference}|} \qquad (12)$$

In our case, the measurement distance is set to 1 m and the reference radar target is a trihedral corner reflector with an edge of 3 cm.

## Data availability
The datasets generated during and/or analyzed in this study are available from the corresponding authors upon reasonable request.

## Code availability
The codes that support this study are available from the corresponding authors upon reasonable request.

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

## Acknowledgements
This work was funded by the Deutsche Forschungsgemeinschaft (DFG, German Research Foundation)—Project-ID 287022738—TRR 196 MARIE within project C09 and by the Internal Grant Agency of Brno University of Technology, project no. FEKT-S-23-8191.

## Author contributions
A.J.S. and P.K. conceived the investigation and initiated collaborations. R.J., N.B., and J.L. established the initial research objectives and J.S.P., A.J.S., P.K., and M.S. contributed to the subsequent development of research questions. J.S.P. designed the 3D-PhC cavities, while P.K. developed the lens. M.S. manufactured the structures. J.S.P. performed the measurements and the estimation of the dielectric properties. J.S.P., A.J.S., P.K., and M.S. contributed to the interpretation of results. R.J., N.B., and J.L. provided funding, space, material resources, and laboratory equipment. J.S.P. organized data, created figures, and took the lead in drafting the manuscript. All authors provided feedback that helped shape the composition of the manuscript.

## Funding

## Competing interests
The authors declare no competing interests.
