## [Peer Review File · Communications Engineering]

Reviewers' comments:

Reviewer #1 (Remarks to the Author):

This paper demonstrated a wireless passive sensor at W band using 3-D printed Al₂O₃ EBG structure to measure temperatures >1000 degree C. The fundamental theory is that the dielectric constant of Al₂O₃ increases vs. temperature, and therefore, the resonant frequency of the defect inside the EBG decreases vs temperature. Since Al₂O₃ is a high-temperature-stable dielectric material, this sensor is able to measure at temperatures above 1000 degree C. One more contribution from the authors is that a 3-D printed flatten lens is integrated on top of the EBG structure to provide much higher gain, ultimately leading to a larger reading distance, i.e. 50 cm in this paper.

This paper is well written, the details regarding design, fabrication and measurement are thoroughly described. The measurement results closely match the simulations and support the claims from the authors.

However, I have a few comments below:

1. The dimensions of the sensor are not small though it operates in W band due to the use of EBG structure. The added lens makes it even thicker. Which applications are this type of sensors for? In many applications at such high temperatures, size (particularly thickness) is very critical.
2. Other researchers demonstrated the wireless passive sensors for high temperature applications such as: H. Cheng, X. Ren, S. Ebadi, Y. Chen, L. An and X. Gong, "Wireless Passive Temperature Sensors Using Integrated Cylindrical Resonator/Antenna for Harsh-Environment Applications," in IEEE Sensors Journal, vol. 15, no. 3, pp. 1453-1462, March 2015, doi: 10.1109/JSEN.2014.2363426. Even though that work was demonstrated at around 10 GHz, the sensor operating principle and sensing mechanism (send out a wide-band signal and use the time domain gating to isolate the high-Q resonance from the scattering noise from the environment) are very similar. Authors should not exclude this paper or other related works in the references. Also noted is that many references are very old (before 2000). The authors should do a thorough research on related articles in the past 10 years to reflect the state of the art of this research.

Reviewer #2 (Remarks to the Author):

In manuscript no. COMMS-23-0468-T, authors have demonstrated an alumina-based photonic crystal structure that can be operated at W-band. Depending upon the induced structural defects, this structure excites two high-quality factor resonances ($Q \sim 400$ to 700) at frequencies of 83.85 GHz and 85.50 GHz, respectively at ambient temperature. Authors have utilized this structure to measure high-temperature variation (typically from 0 to 1200 degrees Celsius). The mechanism behind this sensing is the change of the dielectric constant of photonic crystal material (alumina) with temperature. Such variation would tune the band gap and resonating frequencies of photonic crystals. The authors have fabricated their design using Lithography-based ceramic manufacturing technology and characterized using a vector network analyzer (Keysight technology) and W-band horn antenna. Authors have also used the flattened Luneburg lens to increase the readout of wireless signals. Overall, the paper manuscript is written well, and the arguments are justified. Nevertheless, this manuscript requires thorough clarification on the following points:

1. This manuscript is based on the fact that temperature sensing is implemented by the detection of two high Q resonances. However, it's not clear from the manuscript, how two resonances provide superiority in sensing, in comparison with single resonance. For instance, in Figure 4, the authors have demonstrated the performance of their proposed structure, where both the resonances show almost equivalent frequency tunability (Figure 4a). Moreover, S_{11} parameters and Q -factors of resonances are not very divergent (Figures 4b, 4c). Extracted parameters using the resonances are also equivalent. Given such a scenario, what's the added advantage(s) of employing two resonances and a larger bandwidth? It should be evident that information of only one resonance can also provide comparable temperature detection ability.
2. Another important concern in this manuscript is the choice of structure or geometry for the sensor. First, the authors have used photonic crystals where the frequency is dependent on the bandgap. How do authors calculate bandgaps for their structure? Its mathematical foundations are missing in this manuscript. Second, have authors optimized their structure? Can authors provide enough evidence that other near-similar geometries or configurations don't provide high-quality factor resonances at similar W-band ranges? Third, how authors compare their performance compared to dielectric metasurfaces, given that both of them provide high- Q resonances (Nature Nanotechnology 11, no. 1 (2016): 23-36; Advanced Materials 31, no. 37 (2019): 1901921; Science Bulletin 64, no. 12 (2019): 836-842; etc.).
3. A concern regarding this sensor is the choice of materials and sensing performance. This crystal is based on alumina. First, the authors should justify their consideration by comparing sensing performance with other dielectric materials and temperature-sensitive materials (at least either from literature or by theory/ simulation). Second, authors should calculate the sensitivity of their structure and compare them with other existent temperature sensors available in the literature and/ or market. Third, the authors should provide an outlook to the prospective readers about the limitations of such structures, for example, the temperature resolution of the sensor, temperature detection range, and the effect of external ambiances on sensing characteristics. Fourth, does the sensor work at any medium? How does the performance of photonic crystal compare if it is put to any hot liquid and hot air, given that their dielectric properties should also tune the photonic band gap?

4. A few minor comments also needed to be addressed. First, on page 3, can the authors explain how reflection dip becomes reflection peak after performing time windowing? It seems like authors intentionally getting rid of some useful information for their work. Is this the fact? Second, In Figure 2, responses with and without the lens are confusing. Is the lens also a part of the photonic crystal (which should not be)? Given the experimental data does not clarify (in Figure 2d), can authors at least supplement them with simulations/ theory? Third, given this structure is also sensitive to the ambiance, could authors also provide corresponding received signals solely from the ambiance (without PhC) in Figure 3d? In Figure 4, high-temperature responses are largely divergent. Authors should also provide corresponding variance values with corresponding plots in Figure 4.

Reviewer #1 (Remarks to the Author):

This paper demonstrated a wireless passive sensor at W band using 3-D printed Al₂O₃ EBG structure to measure temperatures >1000 degree C. The fundamental theory is that the dielectric constant of Al₂O₃ increases vs. temperature, and therefore, the resonant frequency of the defect inside the EBG decreases vs temperature. Since Al₂O₃ is a high-temperature-stable dielectric material, this sensor is able to measure at temperatures above 1000 degree C. One more contribution from the authors is that a 3-D printed flatten lens is integrated on top of the EBG structure to provide much higher gain, ultimately leading to a larger reading distance, i.e. 50 cm in this paper.

This paper is well written, the details regarding design, fabrication and measurement are thoroughly described. The measurement results closely match the simulations and support the claims from the authors.

However, I have a few comments below:

Dear reviewer 1, thank you for your time, feedback and constructive criticism. Please find below a point-to-point answer to your comments.

1. The dimensions of the sensor are not small though it operates in W band due to the use of EBG structure. The added lens makes it even thicker. Which applications are this type of sensors for? In many applications at such high temperatures, size (particularly thickness) is very critical.

Authors' comment: We acknowledge that sensor thickness is a critical parameter, as thinner sensors (lower contact surface / volume ratio) lead to a more homogenous and quicker heating or cooling, decreasing temperature gradients in the structure that would lead to larger measurement deviations. In our case, the porosity of the sensor (from the perspective that there are multiple gaps in the 3D PhC structure that allow warm/cold air to seep in) should reduce the response time, although this has not been compared with a thick alumina block. However, as the application is focused on enabling sensing in harsh environments above 75 GHz, we focused on the wireless readout and high-temperature response. Regardless, we have calculated the time constant of the sensor, by profiting from the fact that we hold the temperature of the furnace during 30 min at 600 °C, 700 °C ... 1200 °C. We employed the difference between that moment and the time when the f_{res} stabilizes (i.e., $df_{res}/dt \approx 0$), which we consider that it corresponds to when the sensor has reached the target temperature. By doing so, we have estimated a time constant τ of approximately 1.5 min. We have included this information in a new section: "Limitations".

The main application for which this sensor is proposed is as a chipless RFID tag within a cooperative indoor localization infrastructure. Indoor localization and mapping based on chipless RFID tags in the mm-wave band / sub-THz band is a growing interesting topic, as this frequency band combines the higher localization accuracy of optical methods with the capability to penetrate trough materials and gases of the microwave regime [A, B, C]. Particularly in the case of indoor hazardous environments (like a fire) such system could be employed to localize human beings in buildings covered by smoke, estimate the status of the building, etc.

[A] El-Absi, M. et al. "High-Accuracy Indoor Localization Based on Chipless RFID Systems at THz Band". IEEE Access 654355–54368 (2018).

[B] Jiménez-Sáez, A. et al. "Frequency-Coded mm-Wave Tags for Self-Localization System Using Dielectric Resonators". Journal of Infrared, Millimeter, and Terahertz Waves 41 (8) (2020).

[C] El-Absi, M. et al. "Chipless RFID Tags Placement Optimization as Infrastructure for Maximal Localization Coverage". IEEE Journal of Radio Frequency Identification 6, 368–380 (2022).

Authors' action: We have added a paragraph (pp. 2, 1st col., lines 59-73) in the introduction describing the potential application of this sensor. Further, we address the thickness issue in a new section of the manuscript, "Limitations" (pp. 7-8).

2.1. Other researchers demonstrated the wireless passive sensors for high temperature applications such as: H. Cheng, X. Ren, S. Ebadi, Y. Chen, L. An and X. Gong, "Wireless Passive Temperature Sensors Using Integrated Cylindrical Resonator/Antenna for Harsh-Environment Applications," in IEEE Sensors Journal, vol. 15, no. 3, pp. 1453-1462, March 2015, doi: 10.1109/JSEN.2014.2363426. Even though that work was demonstrated at around 10 GHz, the sensor operating principle and sensing mechanism (send out a wide-band signal and use the time domain gating to isolate the high-Q resonance from the scattering noise from the environment) are very similar. Authors should not exclude this paper or other related works in the references.

Authors' comment: At the beginning we decided to discard several publications because they operated at relatively low frequencies compared to the W-band. However, it is right that they should be referenced in the manuscript, as the approach followed for sensing is the same.

Authors' action: We have added a brief mention in the introduction (pp. 2, 2nd col., lines 61-64) referencing ~10 GHz chipless sensors (highest frequency that we found) operating beyond 1000 °C, and a table (pp. 8) comparing them in the conclusion of the manuscript.

2.2. Also noted is that many references are very old (before 2000). The authors should do a thorough research on related articles in the past 10 years to reflect the state of the art of this research.

Authors' comment: We did an extensive literature review and, to the best of our knowledge, found a publication gap for wireless temperature sensors operating in the W-band beyond 1000 °C, which led us to establish the equivalence with material characterization systems. Even in this area, publications that deal with this frequency range at these temperatures are scarce, as noted by how old the references are. Although we did not find newer publications, we will be grateful to know if we missed them and are open to add them to the manuscript.

Authors' action: We have remarked this, to the best of our knowledge, lack of publications dealing with wireless temperature sensors at W-band and operating up to and beyond 1000 °C (pp. 2, 1st col., lines 74 – 82).

Reviewer #2 (Remarks to the Author):

In manuscript no. COMMS-23-0468-T, authors have demonstrated an alumina-based photonic crystal structure that can be operated at W-band. Depending upon the induced structural defects, this structure excites two high-quality factor resonances ($Q \sim 400$ to 700) at frequencies of 83.85 GHz and 85.50 GHz, respectively at ambient temperature. Authors have utilized this structure to measure high-temperature variation (typically from 0 to 1200 degrees Celsius). The mechanism behind this sensing is the change of the dielectric constant of photonic crystal material (alumina) with temperature. Such variation would tune the band gap and resonating frequencies of photonic crystals. The authors have fabricated their design using Lithography-based ceramic manufacturing technology and characterized using a vector network analyzer (Keysight technology) and W-band horn antenna. Authors have also used the flattened Luneburg lens to increase the readout of wireless signals. Overall, the paper manuscript is written well, and the arguments are justified. Nevertheless, this manuscript requires thorough clarification on the following points:

Dear reviewer 2, we appreciate the time you dedicated to providing valuable feedback and sharing constructive criticism. Please find below a detailed response addressing each of your comments.

1. This manuscript is based on the fact that temperature sensing is implemented by the detection of two high Q resonances. However, it's not clear from the manuscript, how two resonances provide superiority in sensing, in comparison with single resonance. For instance, in Figure 4, the authors have demonstrated the performance of their proposed structure, where both the resonances show almost equivalent frequency tunability (Figure 4a). Moreover, S11 parameters and Q-factors of resonances are not very divergent (Figures 4b, 4c). Extracted parameters using the resonances are also equivalent. Given such a scenario, what's the added advantage(s) of employing two resonances and a larger bandwidth? It should be evident that information of only one resonance can also provide comparable temperature detection ability.

Authors' comment: We agree that employing one resonance is sufficient to realize a temperature sensor. However, we employed two for mainly two reasons (i) from the material characterization point of view, it gives redundancy to the results. (ii) from the sensing perspective, the addition of a second cavity allows for employing the separation between their resonance frequencies, $\Delta f = |f_{res,2} - f_{res,1}|$, to encode different sensors. In a high-temperature scenario where several sensors are deployed and only one resonance is considered, the shift on $f_{res} \propto T$ can lead to overlapping between the responses of several sensors. By designing each of them with a specific Δf , a distinction, i.e., identification, becomes feasible. This could also be achieved if each sensor is assigned a "frequency slot" broad enough to contain their shift of the response regarding temperature, but it is a less efficient allocation of the spectrum.

Authors' action: We have updated the manuscript with this information, in pp. 3, 2nd col., lines 136 - 141.

2.1. Another important concern in this manuscript is the choice of structure or geometry for the sensor. First, the authors have used photonic crystals where the frequency is dependent on the bandgap. How do authors calculate bandgaps for their structure? Its mathematical foundations are missing in this manuscript.

Authors' comment: As mentioned in the publication (pp. 3, 2nd col., lines 147-153), the design is based in the results presented by Povinelli et al. [Ref. 38]. The paper presents the band structure of the 3D PhC that we implemented, with the difference that their band structure is calculated for a relative permittivity contrast of 12:1. Hence, we selected that the central frequency of our PhC should be at 87.5 GHz (centre of the W-band), so, following that publication, the nearest-neighbour spacing, a , can be calculated as

$$\frac{\omega a}{2\pi c} = 0.4 \rightarrow a(12:1) = 1.37 \text{ mm}$$

In this case, since we assumed that the ϵ_r from Alumina is 9.5, we applied the scaling properties of the Maxwell equations to scale up a to the corresponding value

$$a(9.5:1) = \sqrt{\frac{12}{9.5}} \cdot a(12:1) = 1.54 \text{ mm}$$

Then we adjusted the values of a slightly by simulating the reflection coefficient of the PhC with the time-domain solver of CST Studio Suite. We have included a section in Methods, "Estimation of the lattice constant", summarizing this process.

Authors' action: Added a section in Methods, "Estimation of the lattice constant" (pp. 10, 2nd col., lines 482 - 502), summarizing the described process.

2.2. Second, have authors optimized their structure? Can authors provide enough evidence that other near-similar geometries or configurations don't provide high-quality factor resonances at similar W-band ranges?

Authors' comment: Other geometries can absolutely achieve high-Q cavities at the W-band, even higher than the ones presented in our publication, provided that they can be fabricated with the corresponding dimensions.

The optimization performed in the structure is to adjust slightly the limits of the bandgap, as aforementioned in Q2.1. In terms of high-Q structures, there are multiple designs and geometries that can achieve high-Q factors. For example, cavities within 2D PhC can be optimized to achieve virtually any desired Q-factor [A] or can be stacked [B]. Bragg reflectors can be designed to present high-Q factors too, as well as the structures in the works suggested in Q2.3. At the end, the main limiting factor with such structures to achieve a high loaded Q-factor will be the material losses [C].

We propose the employment of cavities embedded within 3D PhC not only for their inherent high-Q factors, but also because (i) the 3D PhC by itself prevents the cavities begin affected by its surrounding media, due to the 3-dimensional EBG, with some constrains (please refer to Q3.3 for more details) (ii) it is a mechanically stable structure and (iii) its implementation in Al_2O_3 allows for its manufacturing with Lithography-based Ceramic Manufacturing technology (3D printing), enabling its future integration with the flattened lens as a monolithic block.

- [A] D. Englund, I. Fushman, and J. Vuckovic, "General recipe for designing photonic crystal cavities," *Opt. Express* 13, 5961-5975 (2005).
- [B] Masamichi Ito et al., "Enhancement of Cavity-Q in a Quasi-Three-Dimensional Photonic Crystal," *Jpn. J. Appl. Phys.* 43 1990 (2004).
- [C] Tao Xu et al., "The influence of material absorption on the quality factor of photonic crystal cavities," *Opt. Express* 17, 8343-8348 (2009).

Authors' action: Connected with Q2.3, we have briefly mentioned the possibility of employing other kind of dielectric metasurfaces that implement high-Q resonances in the manuscript, pp. 3, 2nd col., lines 122 – 130.

2.3. Third, how authors compare their performance compared to dielectric metasurfaces, given that both of them provide high-Q resonances (*Nature Nanotechnology* 11, no. 1 (2016): 23-36; *Advanced Materials* 31, no. 37 (2019): 1901921; *Science Bulletin* 64, no. 12 (2019): 836-842; etc.).

Authors' comment: Thank you for the very interesting references. Do note that all of them are focused in frequencies > 100 THz. However, it is true that they can be scaled down to the microwave regime. Most interestingly, some of them have already been employed to design temperature sensors and are

cited in our manuscript. For example, Kubina [Ref. 35], Le Floch [Ref. 36] and Cheng [Refs. 29-30] employ high-index dielectric resonators for sensing.

The sensing part of our structure (the PhC) can be summarized in a perfect dielectric reflector, that implements frequency coding at a specific frequencies. Any dielectric metasurface that is able to do so, either by

- (i) implementing absorption of the resonances,
- (ii) employing retroreflective cavities (as it is our case),
- (iii) allowing only specific frequency components to propagate through it while reflecting the rest (transmission filter)

and can be fabricated with ceramics (suitable for harsh environments) can be employed as the sensing part. However, as elaborated in Q4.1, we prefer the employment of retroreflective high-Q cavities, whose resonance frequency can be readout as a peak in the frequency domain for temperature sensing. That does not imply that structures with a notch in the frequency domain cannot be employed as temperature sensors, as demonstrated in Refs. 33-34, although the successful sensor readout will be more vulnerable to the presence of clutter.

Authors' action: We have briefly mentioned the possibility of employing other kind of dielectric metasurfaces that implement high-Q resonances in the manuscript, pp. 3, 2nd col., lines 122 – 130.

3.1. A concern regarding this sensor is the choice of materials and sensing performance. This crystal is based on alumina. First, the authors should justify their consideration by comparing sensing performance with other dielectric materials and temperature-sensitive materials (at least either from literature or by theory/ simulation).

Authors' comment: Alumina benefits from several advantages that made it a particularly useful material for our sensor, besides of its temperature sensitivity in terms of resonance frequency shift, which is around -60 ppm/K.

First, it can be 3D-printed via Lithography-based Ceramic Manufacturing, which is useful to manufacture complex structures, such as the lens, the cavities, and the PhC (While the PhC itself is feasible to be manufactured with several processes such as drilling, like Yablonovitch et al. did in the first publication dealing with these type of PhCs [D], the cavities inside of it are more challenging to achieve).

Second, it presents very low dielectric losses in the millimetre wave spectrum, of the order of $4 \cdot 10^{-4}$ at ambient temperature, which is important for both the implementation of high-Q cavities and achieving long range backscattering of their responses.

Third, it presents a ϵ_r of ~ 9 , which allows for the implementation of the lens (which requires a $\epsilon_{r,eff} \sim 1$ in its forefront) and eases the design of high-Q cavities. Please do note that in Q2.2. we mentioned that high-Q is feasible for several structures. Cavities implemented in a 3D PhC made from Alumina easily achieve high-Q factors, due to the moderately high ϵ_r , the low dielectric losses and the 3D EBG.

[D] Yablonovitch et al., "Photonic band structure: The face-centered-cubic case employing nonspherical atoms". Phys. Rev. Lett. 67, 2295

Authors' action: We added a paragraph in the introduction (pp. 2, 2nd col., lines 74-78) justifying the employment of Alumina as the material for the temperature sensor.

3.2. Second, authors should calculate the sensitivity of their structure and compare them with other existent temperature sensors available in the literature and/ or market.

Authors' comment: We define the sensitivity as the shift in the resonance frequency in ppm/K from room temperature to the maximum measured temperature, which for our sensor is of -57 ppm/K. We have included a table comparing several wireless microwave sensors operating around 1000 °C, from which it can be seen that Alumina-based sensors present a large sensitivity compared to the others.

Authors' action: We have added a table comparing different sensors in terms of sensitivity in the conclusion of the manuscript (pp. 8).

3.3. Third, the authors should provide an outlook to the prospective readers about the limitations of such structures, for example, the temperature resolution of the sensor, temperature detection range, and the effect of external ambiances on sensing characteristics.

Authors' comment: We agree that it is beneficial to include more detailed information about the limitations.

Authors' action: We have added a new section, "Limitations", to the manuscript, pp. 7-8.

3.4. Fourth, does the sensor work at any medium? How does the performance of photonic crystal compare if it is put to any hot liquid and hot air, given that their dielectric properties should also tune the photonic band gap?

Authors' comment: The performance of the sensor will change depending on the surrounding environment. In the case of hot air, we do not expect any issue, as it was measured in that situation. However, it bears to mention that changes in the surrounding media will affect in two ways.

On the one hand, a change in the surrounding media will definitely tune both the bandgap and the cavities' resonance frequencies, shifting them towards lower frequencies. There are two possibilities to account for this: either a reader operating at lower frequencies is employed, or the PhC is designed with knowledge of in which environment is going to be employed, so its dimensions can be adjusted depending on the expected frequency shift. It must be mentioned that this would also require a re-design of the lens, as the designed gradient index is based in the assumption of having a unit cell composed by a mix of Alumina and air.

On the other hand, another challenge arises for lossy media, related to the high-Q cavities. The presence of a lossy environment decreases the Q_t of the cavity (since it can be considered as an increase in the dielectric losses). If this decrease is large enough, then the cavity response will not be appreciable anymore.

Authors' action: We have added a new section, "Limitations", to the manuscript, pp. 7-8.

4.1. A few minor comments also needed to be addressed. First, on page 3, can the authors explain how reflection dip becomes reflection peak after performing time windowing? It seems like authors intentionally getting rid of some useful information for their work. Is this the fact?

Authors' comment: Regarding how the reflection dip becomes a peak after time windowing, it is a consequence of isolating the backscattered response of the cavities in time domain. The cavities are high-Q, which implies that they slowly re-radiate the energy contained within them, long enough to outlive clutter echoes. If there were no cavities, then applying windowing after the reception of the environment echoes leads to noise. However, since the cavities are there, they can be considered as sources radiating at their resonance frequency, hence receiving only their responses, which are seen as a peak in the frequency domain. Without time gating the reflection from the PhC adds to the cavities' reradiated power, resulting in a dip or notch in the received frequency-domain spectrum.

Regarding on whether we are getting rid of useful information: we acknowledge that the EBG could be used to extract information from the sensor, such as in [E]. However, in our work, we are assuming the presence of strong clutter near the sensor, which might mask the first echo from the EBG, leading to an unsuccessful detection.

Furthermore, do note that the notch depth is also proportional to the unloaded (Q_u) and external (Q_e) Q-factors of the cavity [F], so that it is possible to have a reflection dip with a very small depth regarding its surrounding spectrum, while still presenting a high magnitude peak after time-gating is applied. This is appreciable in the manuscript, where the notch depth before isolating the cavities' responses is different for Figs. 2b, 2d and 3c. The difference between Figs. 2d and 3c is due to the manual positioning of the lens on top of the PhC for each measurement, which leads to a different Q_e for each case. In the case of Fig. 2b, it corresponds to the measurement of the PhC without lens. This effect also has a direct consequence for the sensor at different temperatures. As shown in our results, the dielectric losses increase with increasing temperature, which leads to a decrease in Q_u and thus to variations in the notch depth with temperature.

In light of the above, we consider that it is more reliable to perform time-gating and employ the cavities' isolated backscattered responses (peaks) to extract the information that we need, in order to account for both strong clutter and variations in the notch depth.

- [E] S. P. Hehenberger et al., "Broadband Effective Permittivity Simulation and Measurement Techniques for 3-D-Printed Dielectric Crystals," in IEEE Transactions on Microwave Theory and Techniques, vol. 71, no. 10, pp. 4161-4172, Oct. 2023.
- [F] Che Qu et al. "Tailor the Functionalities of Metasurfaces Based on a Complete Phase Diagram," Phys. Rev. Lett. 115, 235503 (2015).

Authors' action: We have provided a deeper explanation to the reviewer.

4.2. Second, In Figure 2, responses with and without the lens are confusing. Is the lens also a part of the photonic crystal (which should not be)? Given the experimental data does not clarify (in Figure 2d), can authors at least supplement them with simulations/ theory?

Authors' comment: It is difficult to simulate the scenarios considered in Fig. 2d, due to the large amount of computational resources that are needed, which is outside of what our workstations can handle. However, we acknowledge that Fig. 2 was confusing. We have tried to clarify it by adding the different structures that are being measured, along with their measurement set-up, as well as separating the responses for each structure. We hope the figure is more understandable now.

Authors' action: We have made important modifications to Fig. 2 to clarify the data being presented in each subfigure.

4.3. Third, given this structure is also sensitive to the ambiance, could authors also provide corresponding received signals solely from the ambiance (without PhC) in Figure 3d? In Figure 4, high-temperature responses are largely divergent. Authors should also provide corresponding variance values with corresponding plots in Figure 4.

Authors' comment: Regarding Fig. 3, we have measured the response of the empty furnace for all temperatures considered in the work and added it to the graphs. As expected, the employment of time-gating after the furnace's echoes have decayed allows for the correct identification of the two cavity peaks, due to their high-Q and retroreflective properties.

Regarding Fig. 4, the shadowed plots indicate the 95 % interval for the data (i.e., 2 x standard deviation). Since the variance and standard deviation are related, we decided to provide the former one. Regardless, we understand that the clutter present in the plot might lead to confusion about this. We have worked it further, by slightly adjusting the markers.

Furthermore, we have generated the following alternative image, where the whiskers denote the 95 % interval. However, we believe that the current Fig. 4 allows for the quick identification of the higher spread of the results present at high temperature, consequence of the increased dielectric losses. Regardless, we are open to switch the images if the reviewers consider it necessary for the improvement of the final manuscript.

Figure 1. Revised Fig. 4 with 95 % interval marked with whiskers. Translucent markers correspond to the data points, while solid ones to the median value.

Authors' action: We have prepared alternative images and explained our reasons for keeping the original one. Slight modifications were applied to Figs. 3 and 4 in the revised manuscript.

Best wishes!

Thank you :)

Reviewers' comments:

Reviewer #1 (Remarks to the Author):

I am not convinced that the main application of this sensor is as a chipless RFID tag within a cooperative indoor localization infrastructure. Even with 50 cm sensing distance (pretty good compared to other >1000 degree C passive sensors), it is not sufficient for Indoor localization and mapping based on chipless RFID tags.

The authors did mention that "Particularly in the case of indoor hazardous environments (like a fire) such system could be employed to localize human beings in buildings covered by smoke, estimate the status of the building, etc." However, there are much better technologies including active beamers protected by high-temperature materials.

One of the really useful applications for such sensors is to sense temperatures inside turbines since active devices cannot survive the harsh environments in the turbines. But the proposed sensor is too bulky. Also in the rebuttal, the authors mentioned the time constant is ~1.5 minutes which is too slow for such applications.

Reviewer #2 (Remarks to the Author):

Authors have incorporated my recommendations in manuscript, except comment no. 4.1. I would encourage authors to provide this details more elaborately (maybe with some illustrations) as an Appendix or in Supplement.

Reviewer #3 (Remarks to the Author):

The manuscript proposes a wireless 3D-printed sensor for high-temperature applications. The manuscript is not well motivated and is written that better fits a disciplinary journal. For instance, the abstract and introduction provide little context for the need or technical challenges of the proposed efforts/technology for broad audience. In its current form, it does not suit an interdisciplinary journal. Similarly, the context for the very proposed technology is not very clear.

For instance, what is the need to explain the working principle of the sensor without elaborating the technical bottleneck or need. Similarly, why is the sensor 3D-printed (as emphasized in the abstract). Could other techniques be used? Again, it is not clear what the key advancements are. Thus, the manuscript can be improved by better contextualizing their challenges and proposed solutions.

The authors categorize their findings as empirical and the manuscript mainly reports on the conducted work without highlighting the novelty.

The contributions don't seem to be on 3D-printing as a readily developed technology (LCM) seems to have been used. Nevertheless, the aluminum oxide is the material, not the structure. Please revise the Fabrication Section accordingly. Please provide the reasoning behind selected processing parameters (e.g., Ultrasonication time) and how alterations of such parameters can lead to variability in reported results. Also, if a commercial printer has been used, the brand is not reported.

Comments on Figures:

Fig. 2a is not legible

Fig. 2b, 3d, the data is blocked under the legend

Fig. 3a is not very intuitive

Fig. 3b-d, the data suffers from being hidden under captions

Dear reviewers, thank you for your constructive feedback and time dedicated to our work. We hope we have addressed your comments in this revised version and please kindly find a detailed answer to them below. The changes in the manuscript are marked in blue.

Reviewer #1 (Remarks to the Author):

1. I am not convinced that the main application of this sensor is as a chipless RFID tag within a cooperative indoor localization infrastructure. Even with 50 cm sensing distance (pretty good compared to other >1000 degree C passive sensors), it is not sufficient for Indoor localization and mapping based on chipless RFID tags. The authors did mention that "Particularly in the case of indoor hazardous environments (like a fire) such system could be employed to localize human beings in buildings covered by smoke, estimate the status of the building, etc." However, there are much better technologies including active beamers protected by high-temperature materials.

Authors' comment: Thank you for the remark. The sensing distance can be greatly increased by appropriate readers or lenses with improved gain as we have now included in the Supplementary information. However, it is indeed true that any active sensor provides better functionality, and that operation at high-temperatures is possible by proper isolation. Nevertheless, all active technologies rely on semiconductors and are therefore limited in the time-window that they can be continuously exposed to high-temperatures, as operation will end as soon as they reach temperatures of between 150 °C (conventional electronics) and 300 °C SOI-CMOS (<https://www.ims.fraunhofer.de/en/Core-Competence/Smart-Sensor-Systems/Integrated-Sensor-Systems/High-Temperature-Electronics.html>). In our case, the sensor can continuously operate at temperatures of 1200 °C due to the absence of semiconductors (chipless) and common good conductors such as copper (melting temperature: 1084°C) and gold (1063°C) <https://www.onlinemetals.com/en/melting-points> . We believe this property can open new opportunities beyond the indoor localization in fire scenarios that we are currently focused on. One example are turbines, as introduced by the reviewer.

The field of chipless sensors focuses on the amount of information, or the integration of sensors, while often neglecting high-temperature operation, readout range and/or clutter suppression. The presented sensor, although limited when compared to active beamers, is a robust solution if compared to chipless alternatives. The main current limitations (readout range and response time) are acknowledged, and potential improvements (higher RCS and reader gain, or improved A/V ratio) are introduced in the manuscript and further detailed in the supplementary information.

Authors' action: We provided a deeper explanation to the reviewer. Also, we modified the introduction of the manuscript and added Supplementary Section S4, discussing how to increase the readout range of the sensor.

2. One of the really useful applications for such sensors is to sense temperatures inside turbines since active devices cannot survive the harsh environments in the turbines. But the proposed sensor is too bulky. Also in the rebuttal, the authors mentioned the time constant is ~1.5 minutes which is too slow for such applications.

Authors' comment: Following this comment, we have reviewed our measurements and conducted further experiments and found that the heating cycle of our furnace is too slow to accurately measure the time constant τ_t . We thank the reviewer for helping us discovering this issue. Hence, we have removed this value from the main body altogether. Instead we have substituted it by:

1. We have included a theoretical analysis of τ_t in the form of Supplementary Section S5, assuming that there is no heat radiation in the environment.
2. We have conducted a further experiment to show how this value changes in the presence of air currents, where the porosity of the PhC structure (filling factor of approx. 11%) allows for a faster τ_t .

Authors' action: We have added Supplementary Section S5, discussing the time constant of the sensor. Further, the previous value of τ_t has been removed from the main body of the publication, reporting instead on the theoretically calculated value.

Reviewer #2 (Remarks to the Author):

Authors have incorporated my recommendations in manuscript, except comment no. 4.1. I would encourage authors to provide this details more elaborately (maybe with some illustrations) as an Appendix or in Supplement.

Authors' comment: Thank you for the suggestion.

Authors' action: We have added Supplement S1, describing the operating principle of high-Q cavities and why a peak appears at their resonance frequency after their response is isolated in time domain.

Reviewer #3 (Remarks to the Author):

The manuscript proposes a wireless 3D-printed sensor for high-temperature applications.

1. The manuscript is not well motivated and is written that better fits a disciplinary journal. For instance, the abstract and introduction provide little context for the need or technical challenges of the proposed efforts/technology for broad audience. In its current form, it does not suit an interdisciplinary journal. Similarly, the context for the very proposed technology is not very clear. For instance, what is the need to explain the working principle of the sensor without elaborating the technical bottleneck or need.

Authors' comment: We understand that the introduction of the manuscript did not convey the motivation of the work and thus have rewritten it extensively to address why such sensor might be useful for interdisciplinary readers.

Authors' action: We have rewritten significant parts of the introduction and operating principle.

2. similarly, why is the sensor 3D-printed (as emphasized in the abstract). Could other techniques be used? Again, it is not clear what the key advancements are. Thus, the manuscript can be improved by better contextualizing their challenges and proposed solutions.

Authors' comment: The sensor structure is too complex to be manufactured in other technology different than 3D-printing without a significant added cost.

On the one hand, the gradient index (GRIN) lens is composed by a grid of spatially varying cubes, with a minimum thickness of 160 μm . Furthermore, it is implemented in 3D. For such lenses (either made from dielectric materials [A] or metals [B]), the current standard is their manufacturing via 3D printing, due to the flexibility that this technique offers. For example, drilling limits the GRIN implementation to discrete circular steps, and it is employed mainly in planar lenses such as in [C], which is not our case.

On the other hand, 3D photonic crystals operating in the microwave range have been demonstrated by drilling [D], which is not suitable for the millimetre / sub-THz range due to the small sizes involved and because the manufacturing of the cavities is impossible without splitting the structure in two, as they are embedded in the PhC centre. Further, 3D printing, and particularly 3D printing of ceramics, has been proven very suitable for the implementation of such structures [E, Ref. 43].

In light of the above, we decided not to include a detailed analysis of other manufacturing techniques, as we do not see a real alternative to 3D printing for the manufacturing of chipless, monolithic, high-temperature, millimetre wave sensors (lens & PhC together) which follow this approach.

[A] Munina, et al., "A Review of 3D Printed Gradient Refractive Index Lens Antennas," IEEE Access, 2023

[B] Shamvedi, O'Leary and Raghavendra, "Development of 3D Printed Metallic Lenses," EuCAP 2023.

[C] N. C. Garcia and J. D. Chisum, "High-Efficiency, Wideband GRIN Lenses With Intrinsically Matched Unit Cells," IEEE Transactions on Antennas and Propagation, 2020.

[D] Yablonovitch et al., "Photonic band structure: The face-centered-cubic case employing nonspherical atoms".
Phys. Rev. Lett. 67, 2295

[E] Lu et. Al, "Fabrication of Millimeter-Wave Electromagnetic Bandgap Crystals Using Microwave Dielectric Powders," Journal of the American Ceramic Society, 2009

Authors' action: We provided a deeper explanation to the reviewer.

3. The authors categorize their findings as empirical and the manuscript mainly reports on the conducted work without highlighting the novelty.

Authors' comment: Please note that Communications Engineering states in their Style and formatting guide that "*Language such as "new", "novel", "for the first time", "unprecedented", etc, should be avoided because it often leads to unproductive controversy. Novelty should be clear from the context."* Thus, we tried to address that by including Table. I in the manuscript. Furthermore, we have modified the introduction to stress out the main contributions of our work.

Authors' action: We have marked the main contributions of the work at the end in the Introduction section.

4. The contributions don't seem to be on 3D-printing as a readily developed technology (LCM) seems to have been used. Nevertheless, the aluminum oxide is the material, not the structure. Please revise the Fabrication Section accordingly. Please provide the reasoning behind selected processing parameters (e.g., Ultrasonication time) and how alterations of such parameters can lead to variability in reported results. Also, if a commercial printer has been used, the brand is not reported.

Authors' comment: Ultrasonic energy is used to accelerate the removal of unpolymerized slurry from the structures. On the one hand, ultrasonication time should not be too short otherwise cleaning won't be sufficient. On the other hand, ultrasonication for a long period (i.e. exposing the structures to too much ultrasonic energy) may lead to some damage in the structures and should be avoided. Our finding showed that 1-5 minutes is a safe range.

Authors' action: We have amended the Fabrication section as requested by the reviewer, including a short sentence about the selection of parameters important for the printing process.

5. Comments on Figures:

- Fig. 2a is not legible
- Fig. 2b, 3d, the data is blocked under the legend
- Fig. 3a is not eru intuitive
- Fig. 3b-d, the data suffers from being hidden under captions

Authors' comment: Thank you for the comment. We have tried to fit the images and legends so that the data is not obstructed. In the cases where this happens, the part of data partially hidden (As all legends are see-through) are not significant for the outcomes of our work.

Authors' action: We have modified Figs. 2 and 3 as proposed by the reviewer.

REVIEWERS' COMMENTS:

Reviewer #1 (Remarks to the Author):

The authors explained the limitation of sensing range and offered potential ways to enhance the range.

In addition, the authors revised the section related to the time constant.

However, I am still not convinced how the proposed sensors can be practical to use though the technical presentations are valid and complete.

Reviewer #2 (Remarks to the Author):

No other comments. Good luck!

Reviewer #3 (Remarks to the Author):

I don't have any further comments.

Dear reviewers, we thank you for your constructive feedback and time dedicated to our work.

Reviewer #1 (Remarks to the Author):

The authors explained the limitation of sensing range and offered potential ways to enhance the range.

In addition, the authors revised the section related to the time constant.

However, I am still not convinced how the proposed sensors can be practical to use though the technical presentations are valid and complete.

Reviewer #2 (Remarks to the Author):

No other comments. Good luck!

Reviewer #3 (Remarks to the Author):

I don't have any further comments.